# A ShExML perspective on mapping challenges: already solved ones, language modifications and future required actions

Herminio García-González

IT and Communications Service, University of Oviedo, Asturias, Spain
`garciaherminio@uniovi.es`

**Abstract.** Data mapping languages allow users to create knowledge graphs with lower cost and time. Some challenges cannot be solved with state-of-the-art languages and tools, though. Thus, in this paper we use and modify ShExML to deal with some of them. We see how some challenges were already solved, which modifications we had to perform to cover others, and how the rest of them could be covered in future versions. Then, we establish a demonstration on language integrity after changes and a discussion on performed and upcoming changes. These solutions, alongside the discussion and join analysis of other languages and tools solutions, will drive us to effective techniques to solve all proposed challenges.

**Keywords:** mapping challenges, ShExML, data mapping languages, knowledge graph construction

## 1 Introduction

Mapping heterogeneous datasources using a single representation is an active field which has been getting traction in the past years. For this purpose a set of languages and tools have been proposed [7] which lower the cost and time employed in these tasks. This trajectory ended up in the celebration of the 1st International Workshop on Knowledge Graph Building [1] and, one year after, the beginning of the Knowledge Graph Construction W3C Community Group[1] where academia, industry and practitioners are gathered to envision new steps, find unsolved problems, and face new challenges in this field[2]. One of the community outputs was a set of mapping challenges[3] which are nowadays complicate to solve with the current state-of-the-art languages, tools and techniques.

Therefore, in this paper we tackle some of these mapping challenges with ShExML [4] and try to solve the following research questions:

– RQ1: How can mapping challenges be solved with ShExML?

---

[1] https://kg-construct.github.io/tpac-web/
[2] https://kg-construct.github.io/tpac-web/#report
[3] https://kg-construct.github.io/workshop/2021/challenges.html

– RQ2: How can not addressed challenges be solved and implemented in ShExML?
– RQ3: Have modifications in ShExML affected the functioning of already present features?

The rest of the paper is structured as follows: in Section 3 we see how current language specification and engine can solve some of the challenges, we explain how solutions for other challenges have been implemented and included in ShExML in Section 4. In Section 5 we propose some further language modifications and a discussion on how the rest of the challenges could be addressed. We demonstrate old features integrity after including new ones and we establish a discussion on mapping challenges results in Section 6. And, finally, in Section 7 we draw some conclusions.

## 2    Mapping challenges summary

During consecutive meetings in the Knowledge Graph Construction W3C Community Group, several mapping challenges and problems were arisen which are collected in the workshop website[4]. Thus, in this paper we deal with this selection of mapping challenges, we examine them and propose solutions within the ShExML language and our engine[5]. To categorise these solutions we link them to ShExML versions so we can trace when these solutions were achieved, i.e., if they were possible to solve before mapping challenges were defined (ShExML v0.2.3), if they were solved after mapping challenges were defined (ShExML v0.2.4 & v0.2.5) or if they are not yet solved (future versions).

Thus, in Table 1 a summary table is offered with the addressed challenges and with which version of ShExML engine the expected output is achieved. Besides, we offer a webpage[6] with links to working solutions as supplementary material for the sake of demonstration and reproducibility.

In the following sections we explain how solutions are achieved, which ShExML constructions and techniques were used, we establish a discussion on reached solutions and how not solved challenges could be addressed in ShExML.

## 3    Already solved mapping challenges

In this section, we deal with mapping challenges that can be solved without any modification in ShExML language and engine. Therefore, these solutions are those which are reachable with ShExML v0.2.3 (released on 29th October 2020)[7], that is, before mapping challenges were defined.

---

[4] https://kg-construct.github.io/workshop/challenges.html
[5] https://github.com/herminiogg/ShExML
[6] http://herminiogg.github.io/mapping-challenges/challenges/solutions.html
[7] https://github.com/herminiogg/ShExML/releases/tag/v0.2.3

| | Already solved (v0.2.3) | With language modifications (v0.2.4 & v0.2.5) | All challenges solved |
|---|---|---|---|
| **Access fields outside iterators** | x | ✓(input 1) | x (input 2) |
| **Datatype map** | ✓(input 5) | ✓ | ✓ |
| **Excel style** | x | x | x |
| **Generate multiple values** | x | ✓ | ✓ |
| **Join on literal** | ✓ | ✓ | ✓ |
| **Language map** | x | ✓ | ✓ |
| **Multivalue references** | x (bug) | ✓ | ✓ |
| **Process multivalue references** | x | x | x |
| **RDF Collections** | x | ✓(input 1) | x (input 2 & 3) |

**Table 1.** Coverage summary table of mapping challenges in ShExML language and engine. ✓means covered or partially covered and **x** means not covered or not completely covered.

### 3.1   Datatype map (input 5)

Datatype map refers to the possibility to generate datatype tags from the input content. Therefore, instead of defining them statically in the mapping rules, this challenge aims to support the dynamic generation of datatype tags from input content. In the case of input 5, it is intended that mapping languages would be able to generate datatype tags according to the most probable value according to values formats. For example, in input 5 it is expected that number 3 would have an `xsd:integer` datatype and 3.14 would have an `xsd:decimal` one.

This inference mechanism was already implemented in ShExML engine which in case that the user does not specifically define a datatype for an object value it will infer the most probable one (see Listing 1.1). Although the current implementation solves this specific mapping challenge, it is a naïve implementation. However, it can lead to a more complex inference system if it is desired or needed.

**Listing 1.1.** ShExML datatype inference function.

```
protected def searchForXSDType(o: String): RDFDatatype = {
  if(Try(o.toInt).isSuccess)
    XSDDatatype.XSDinteger
  else if(Try(o.toDouble).isSuccess)
    XSDDatatype.XSDdecimal
  else if(Try(o.toBoolean).isSuccess)
    XSDDatatype.XSDboolean
  else
    XSDDatatype.XSDstring
}
```

### 3.2   Join on literal

This challenge refers to the possibility to generate literals from a join condition (i.e., from another source) where R2RML[8] and RML [2] output a resource by default.

In ShExML join conditions generate values without any specific form, so it is not determined in this step if it is a literal or a resource. It is, then, defined by the user in the shapes part where the user decides the form of the output. This is a design decision on ShExML that was driven by the separation of concerns main principle. In Listing 1.2 we can see how the join condition is defined in `familyName` expression, and how then this expression is used in `:Author` shape without any prefix, indicating that a literal must be generated.

**Listing 1.2.** ShExML solution for join on literals.

```
PREFIX : <http://example.com/>
PREFIX experson: <http://example.com/person/>
PREFIX dbr: <http://dbpedia.org/resource/>
PREFIX schema: <http://schema.org/>
SOURCE jsonfile <https://raw.githubusercontent.com/
kg-construct/mapping-challenges/
2aac9680cd731fd647abd33d44a7f400e4278cf3/
challenges/join-on-literal/input-1/input.json>
ITERATOR author <jsonpath: $.author[*]> {
  FIELD id <id>
  FIELD firstname <firstname>
  FIELD affiliation <affiliation>
}
ITERATOR people <jsonpath: $.people[*]> {
  FIELD firstname <firstname>
  FIELD familyname <familyName>
}
EXPRESSION authors <jsonfile.author UNION jsonfile.people>
EXPRESSION familyName <jsonfile.people.familyname UNION
  jsonfile.author.firstname JOIN jsonfile.people.firstname>

:Author experson:[authors.id] {
  :affiliation [authors.affiliation] ;
  :lastName [familyName] ;
}
```

### 3.3   Multivalue references

This challenge deals with the expected output of a hierarchical document (e.g., XML or JSON files) where multiple iterators are used. The discussion in this challenge is whether we produce the cartesian product and provide a join condition to correlate values or if we just translate the hierarchical information as it

---

[8] https://www.w3.org/TR/r2rml/

is without the need to provide any join condition[9]. This case comes more complicated if a join condition needs to be provided over a JSON file because of the impossibility to access parent nodes (see Section 4.1 for the specific challenge on this topic). Therefore, it seems that in hierarchical data the expected output should be a verbatim translation.

In ShExML, this was the default behaviour from its inception as in ShExML first versions it only supported XML and JSON files. Besides, we saw it as a more usable manner to define these mapping as usability is the main goal of the language [3]. Therefore, in Listing 1.3 we can see how using iterators and nested iterators we can cover these hierarchical data models. If a user wants to generate the cartesian product it would be as easy as to define different top level iterators. In Table 1 this challenge is marked as not solved in ShExML v0.2.3 due to a bug when using only the root node (`$`) in the top iterator query. However, we include it here as the coverage of this challenge did not require syntax modifications nor new features in ShExML engine, only a bug fix.

**Listing 1.3.** ShExML solution multivalue references.

```
PREFIX ex: <http://example.com/>
PREFIX exLab: <http://example.com/lab/>
PREFIX exArticle: <http://example.com/article/>
PREFIX exAuthor: <http://example.com/author/>
PREFIX exAff: <http://example.com/aff/>
SOURCE lab_file <https://raw.githubusercontent.com/
kg-construct/mapping-challenges/main
/challenges/multivalue-references
/input-1/input.json>
ITERATOR lab <jsonpath: $> {
  FIELD labName <labName>
  ITERATOR articles <articles[*]> {
    FIELD title <title>
    ITERATOR authors <authors[*]> {
      FIELD name <name>
      ITERATOR affiliation <affiliation[*]> {
        FIELD label <label>
      }
    }
  }
}

EXPRESSION labValues <lab_file.lab>

ex:Lab exLab:[labValues.labName] {
  a ex:Lab ;
  ex:hasArticles @ex:Article ;
  ex:hasMembers @ex:Author ;
}
```

---

[9] See discussion about this topic in RMLMapper reference implementation: https://github.com/RMLio/rmlmapper-java/issues/28

```
ex:Article exArticle:[labValues.articles.title] {
  ex:hasAuthor @ex:Author ;
}

ex:Author exAuthor:[labValues.articles.authors.name] {
  ex:hasAffiliation
   exAff:[labValues.articles.authors.affiliation.label] ;
}
```

## 4 Language modifications

In this section, we deal with mapping challenges that needed some modifications in ShExML language and engine. Therefore, these solutions are those which are reachable with ShExML v0.2.4 (released on 18th January 2021)[10] or with ShExML v0.2.5 (released on 27th January 2021)[11], that is, after mapping challenges were defined.

### 4.1 Access fields outside iterators

Sometimes, in hierarchical data models, there is the need to access values outside the iteration pattern. For example, we may need to obtain values which are parents of the current iterated node. When dealing with XML files it does not involve any modification in ShExML, as using XPath queries we are able to access upper nodes with double dot and slash notation (i.e., `../`). However, when dealing with JSON files, this is not possible because of JSONPath not supporting parent access notation[12].

This is a well-known problem in data mapping languages as they use Json-Path to define values accesses. Indeed, in xR2RML [10], authors defined a property called `xrr:pushDown` that takes a value in the hierarchy and pushes it down into their offsprings iterators so it can be available further [11].

Following this experience with xR2RML, we implemented a similar solution in ShExML using `PUSHED_FIELD` and `POPPED_FIELD` keywords. When using `PUSHED_FIELD` keyword ShExML engine saves the value using the name as the identifier for further uses. Then, when `POPPED_FIELD` is used ShExML engine searches for the saved value with an identifier which is equal to that given in the query part (i.e., inside `<` and `>`). Therefore, in Listing 1.4, `id` field is saved and then used in `cars` iterator, so we can establish a relation from the car to the owner.

**Listing 1.4.** ShExML solution for accessing fields outside iterations.

[10] https://github.com/herminiogg/ShExML/releases/tag/v0.2.4

[11] https://github.com/herminiogg/ShExML/releases/tag/v0.2.5

[12] https://goessner.net/articles/JsonPath/

```
ITERATOR records <jsonpath: $.records[*]> {
  PUSHED_FIELD id <id>
  FIELD enteredBy <enteredBy>
  ITERATOR cars <cars[*]> {
    FIELD make <make>
    POPPED_FIELD carOwner <id>
  }
}
```

## 4.2   Datatype map

As we mentioned in Section 3.1, this challenge aims to generate datatype tags dynamically from data content. Therefore, the datatype inputs can appear in multiple ways: full URI, prefix plus datatype, or simply datatype name without prefix.

ShExML v0.2.3 supports the creation of static datatype tags with prefix plus datatype syntax (see Listing 1.5). Therefore, we should derive this syntax and maintain its proven usability [3] but giving dynamic datatype generation possibilities. The natural expansion of this syntax is to include the same object generation expression but also for datatypes and language tags (see Section 4.4). So, the final syntax is prefix plus generation expression (inside square brackets) as we can see in Listing 1.6. Prefix can be optional if data value already contains it (e.g., input 1 and 2) and values can be transformed using Matcher feature[13] to expected XML Schema valid datatypes (e.g., input 4).

**Listing 1.5.** ShExML static datatypes syntax.

```
ex:Person exPerson:[person.firstname] {
  ex:num [person.num] xsd:integer ;
}
```

**Listing 1.6.** ShExML dynamic datatypes syntax.

```
ex:Person exPerson:[person.firstname] {
  ex:num [person.num] xsd:[person.dt] ;
}
```

## 4.3   Generate multiple values

This challenge wants to address the problem of generating various datatypes or language tags for the same subject (e.g., a multi-language value). Once datatype maps (see Section 4.2) and language maps (see Section 4.4) are supported in ShExML it is straightforward as ShExML will generate a triple per value returned from the object expression. Therefore, to generate multi-language values the syntax is like in Listing 1.7 and to generate multi-language values with a default language the syntax is like in Listing 1.8.

---

[13] http://shexml.herminiogarcia.com/spec/#matcher

**Listing 1.7.** ShExML multiple values with language tags.

```
ex:Person exPerson:[person.lastname] {
  ex:name [person.firstname.label] @[person.firstname.lang] ;
}
```

**Listing 1.8.** ShExML multiple values with language tags and with a default language.

```
ex:Person exPerson:[person.firstname] {
  ex:name [person.firstname] @en ;
  ex:name [person.firstname] @[person.lang] ;
}
```

### 4.4  Language map

As with datatype maps in Section 4.2, language map challenge want to address the problem of generating language tags dynamically from input data. In ShExML v0.2.3 language tags were supported statically, that is, it was possible to tag an object expression with a specific language but it would be applied to all values (see Listing 1.9).

We performed a syntax and engine modification, like in datatype maps, to be able to generate language tags with expressions. Final syntax is @ plus generation expression (between square brackets) as it can be seen in Listing 1.10. Here, again, the idea was to preserve usability as the main goal and to make it as simpler as possible. Input 1 tests the generation with a valid tag following BCP47[14], input 2 tests the transformation of a language value to a valid tag (in ShExML this is done using Matchers functionality[15]), and in input 3 how two different sources can be joined to provide language information.

**Listing 1.9.** ShExML static generation of language tags.

```
ex:Person exPerson:[person.firstname] {
  ex:lastName [person.lastname] @en ;
}
```

**Listing 1.10.** ShExML dynamic generation of language tags

```
ex:Person exPerson:[person.firstname] {
  ex:lastName [person.lastname] @[person.lang] ;
}
```

### 4.5  RDF Collections

This challenge puts on the table the necessity for a mechanism to create RDF Collections from some values. Normally, in ShExML, and in other data mapping

---

[14] https://tools.ietf.org/html/bcp47

[15] http://shexml.herminiogarcia.com/spec/#matcher

languages, when an object generation expression returns multiple values multiple triples are generated (see Section 3.3). However, in certain cases it is necessary to encapsulate these values inside a collection (e.g., to preserve order).

This was already explored by some languages (e.g., SPARQL-Generate [6]) which provide some directives to create collections. Therefore, we applied this experience in ShExML to cover RDF Collections and Containers (i.e., Lists, Seqs, Bags and Alts.). Now, it is possible to indicate to the engine that a collection or container should be generated using keyword AS plus the desired collection or container (i.e., RDFList, RDFBag, RDFSeq or RDFAlt). See Listing 1.11 for an example.

**Listing 1.11.** ShExML support for RDF collections and containers.

```
ex:Article exArticle:[labValues.articles.title] {
  a ex:Article ;
  ex:hasAuthors
    exAuthor:[labValues.articles.authors.name AS RDFList] ;
}
```

## 5    Future required actions

In this section we discuss further challenges that are not solved with previously mentioned modifications. These are challenges that would require to rethought some functionality or to include new ones but that would need from a well planned inclusion due to their possible interference with other features.

### 5.1    Access fields outside iterators (input 2)

Although this challenge was already addressed in Section 4.1, only input 1 was completely solved. In the case of input 2, where data is in the same hierarchical level (like it would come from two different files), using join conditions in ShExML only one car is linked to each owner when the expected result was two cars per person. To solve this problem we think of two possible solutions.

First one is to review join condition functionality to check whether something is failing (a bug) or if join condition need to be rethought and reimplemented to cover further challenges.

Another possibility, which is already present in other languages like YARRRML [5], is to provide conditional generation. With conditional content generation we are able to test a condition (e.g., in input 2 for value equality) and generate or not the resulting triple depending on its result.

### 5.2    Excel style

A classic solution when dealing with Excel sheets was to convert them to CSV a then treat them as tables to be processed by data mapping languages. However,

this challenge found this solution not appropriate when the style of the Excel sheet want to be preserved. Two solutions could cover this challenge.

First one is to preprocess Excel sheet and convert it to CSV but adding columns with style information so it can be processed by state-of-the-art tools. However, it would require some preprocessing work which would weaken the goal of low cost and time invested when using data mapping tools.

Second one is to include a specific Excel processor with its own query language which can express not only access to cell but also to cell and text style. Thus, in Java based implementations it can be considered to use Apache POI to process sheets and include some simple query support to retrieve styles.

### 5.3   Process multivalue references

This challenge is very close to multivalue references (see Section 3.3), but in this case multivalues are included all within a string value and separated by commas.

Therefore, here the challenge is not about how to output multivalues or create RDF collections but how to effectively process these multivalues which need some processing. Therefore, this would require some sort of data transformations functions that can be applied to extracted values. Therefore, the most effective way to extend ShExML and enable users to transform data is to provide the possibility to execute transformation functions which can be defined by users.

Data transformation functions have been already explored in RML through the FnO library [8] which provides a set of implementation independent reusable functions [9]. So, one possibility is to support FnO functions inside ShExML. The advantage of this proposal is that it moves all function infrastructure outside ShExML language and engine. Conversely, we add more dependencies to users (which can find it hard to learn), we force them to use a third party environment and we lose control of this part.

Another possibility is to provide an environment to define inline functions like semantic actions in Shape Expressions (ShEx) [12]. Therefore, we can provide a restricted environment where higher order functions could be executed (see Listing 1.12 for an example). The advantages are that there is no need for third party dependencies, it provides a higher flexibility and users do not need to learn another tool. However, it can increase complexity due to the necessity to know about functional programming.

**Listing 1.12.** ShExML support for RDF collections and containers.

```
PREFIX ex: <http://example.com/>
SOURCE lab_file <https://raw.githubusercontent.com/
kg-construct/mapping-challenges/main/challenges/
process-multivalue-references/input-1/input.json>
FUNCTION splitFunction <n => n.split(',')>
ITERATOR lab <jsonpath: $> {
  FIELD labName <labName>
  ITERATOR articles <article> {
    FIELD title <title>
```

```
    FIELD tags <tags>
  }
}

EXPRESSION labValues <lab_file.lab>

ex:Tag ex:[lab.articles.tag WITH splitFunction] {
  ex:label [lab.articles.tag WITH splitFunction] ;
}
```

### 5.4 RDF Collections (input 2 & 3)

Although RDF collections and containers were included in ShExML (see Section 4.5) input 2 and 3 present some particularities. In the case of input 2 the use of different keys would require a more complex query or some sort of parametrisation in executed queries. In input 3 the per row iteration model for CSV files implemented in ShExML does not create collections effectively. Therefore, it would imply a reimplementation of per row iteration model for these cases. However, it could affect overall functionality for CSV files.

## 6 Evaluation and Discussion

In RQ3 we have posed a question about the possible effects that modifications in ShExML could have in already working features. The idea of this research question is to demonstrate validity of RQ1 solutions alongside old features that should still work as expected. This type of testing, known as regression testing, have been included in ShExML from the very beginning[16] so we are able to add new features in ShExML knowing that old features are still working as expected. Thus, every time a new version is released these tests must be executed to validate language and engine integrity. Continuous integration is the perfect tool for this task, as every time that a change is submitted to ShExML repository all tests are executed to verify integrity. In ShExML repository we have configured Travis CI[17] for this task. Therefore, these regression tests in v0.2.4[18] and v0.2.5[19] are telling us that all features are still working as expected, and equally, giving a negative answer to RQ3. So, we can conclude that integrity is held.

In Sections 3 and 4 we have seen how some mapping challenges were already solved in ShExML and how we have made some modifications in ShExML language and engine to deal with others. These two sections give an answer for RQ1. These solutions were designed to maintain ShExML usability [3] using a

---

[16] To see all tests that are executed over ShExML engine
https://github.com/herminiogg/ShExML/tree/master/src/test/scala-2.12/es/weso/shexml

[17] https://travis-ci.org/

[18] https://travis-ci.org/github/herminiogg/ShExML/builds/755033209

[19] https://travis-ci.org/github/herminiogg/ShExML/builds/756419674

similar and continuist syntax, so that users can use these new features with the minimum learning curve possible; in other words, making the smallest modifications in ShExML syntax. In addition, in Section 3 we have highlighted how ShExML design have already given an answer to some challenges, emphasising how ShExML separation of concerns principle can give answer to some of them (e.g., Join on literal).

In Section 5 we have given some intuition on how remaining challenges could be solved, answering to RQ2. They would require harder and more complex modifications; in some cases the modification of an already working mechanism (e.g., inputs 2 and 3 in RDF Collections), the inclusion of a new iteration model and the design of a new query language (e.g., Excel style) or the choice between two different systems (e.g., data transformation functions in process multivalue references). All these inclusions will require a careful study and implementation in the language so they do not affect other features and to select the better option from a usability perspective.

## 7   Conclusions

In this paper we have explored how ShExML can deal with some of the challenges defined in the Knowledge Graph Construction W3C Community Group. We have divided them into challenges already solved by ShExML before their definition, challenges solved by latest versions of ShExML and challenges that are not yet solved for which we have given some notions and intuitions on how ShExML can be modified to cover them. Furthermore, we have demonstrated that the modification of ShExML to cover new challenges has not affected other language and engine features. Therefore, we see this work as a first step on how challenges can be solved and, together with solutions from other languages and the joint discussion, we will be able to offer unified solutions to posed mapping challenges.

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
