# OpenReview forum: "A ShExML perspective on mapping challenges: already solved ones, language modifications and future required actions"
_eswc-conferences.org/ESWC/2021/Workshop/KGCW — KGCW 2021_

### Official Review · ~Ana_Iglesias-Molina2 · 2021-04-14
**Good proposal of applying the KGC mapping challenges to ShExML**

**Rating:** 7
**Confidence:** 4

**Review:**


This paper presents improvements of the ShExML mapping language in line with the mapping challenges proposed in the Knowledge Graph Construction W3C Community Group. The author describes the challenges dividing them in three categories: already implemented before the proposal of the challenges from the Community Group, implemented after the proposal, and not implemented (with descriptions of how they will be integrated in the language).

I find the paper as a relevant contribution, it gives perspective about applying the mapping challenges to other languages that doesn't match the [R2]RML schema. The proposal improves ShExML, in terms of both syntax and implementation, and match the current needs to describe and transform data, although not all of the issues are solved yet. Some details I appreciate are the idea of linking every improvement to a version, the regression testing, and regarding the datatype and language maps, how they are validated. It is overall well written and structured, yet I would like to remark some issues I think need to be addressed.

* In section 2, the resources provided should be described more in detail. I’m especially referring to "Input 1", "Input 2", etc. that are frequently mentioned in subsequent sections and not mentioned in this section, only the reference link is provided with barely a description.
* Section 4.4 should come before section 4.3. I think it makes more sense to describe datatype and language maps before the examples in Listings 1.7 and 1.8. Additionally, it would be nice to see in Section 4.3 some example with datatype maps, not only language maps. Regarding the language map, I think a clarification of whether it is possible to write in the same rule the datatype and language tags is advisable, despite the fact that the language is only applicable when the datatype is a string.
* In section 4.5 it is mentioned that SPARQL-Generate is able to describe RDF Collections, I find it relevant to also cite xR2RML in this matter.

Minor issues and typos:
* Introduction, page 1, line 9: “which are nowadays **complicated** to solve”  complicate -> complicated
* Page 10: “when the style of the Excel sheet **is wanted** to be preserved” want -> is wanted
* General lack of commas, some sentences are too long and without pause

---

### Official Review · ~Pano_Maria1 · 2021-04-16
**Potentially interesting contribution that remains too superficial**

**Rating:** 5
**Confidence:** 4

**Review:**

This paper presents an overview of how ShExML solves, or could solve, the mapping challenges posed by the Knowledge Graph Construction Community Group. The structure of the paper is well-thought out, starting from those challenges that were already covered by ShExML before the challenges were posed, moving to extensions introduced to cover several of the posed challenges, and ending with a discussion on those challenges still open in relation to ShExML.

A strong point of this paper is that it makes its claims easily verifiable, by linking to a supplementary webpage containing an overview of the challenges and runnable demonstrations that show that the engine can indeed, given the extended constructs, solve the mapping challenges as described in the paper. This is superb.

Regrettably, I find the paper is written in such a way that you can only understand parts of it if you are familiar with ShExML, since the author often provides little textual explanation of how the relevant language constructs work. Instead the author refers to ShExML snippets with examples of the applied constructs, leaving the reader to figure it out on their own.

Reading the paper, it is often unclear whether the author is referencing ShExML the language, or the implementation of the engine, or both.
The paper also suffers from a great many grammatical errors throughout, not aiding readability. Of particular note is the amount of missing articles like "the" or "a".

It is not clear to me into which of the paper categories this submission falls. To me, it seems to match the System/Demo papers category most. The specified amount of pages for a System/Demo paper is 4-6, which this paper doubles. I believe the paper could be shortened by replacing section 3 with a reference to the supplementary solutions webpage, since, with the limited explanation of the constructs, I feel it adds little value to the paper. Section 5 could also be shortened, since most of the future actions are described in a speculative manner.

Overall, I'm a bit torn. Although I believe the extensions to ShExML which solve the mapping challenges would be a very interesting contribution to the workshop, I feel this paper still needs significant work done to
* describe devised (or proposed) solutions more clearly in the text,
* shorten the paper to within the bounds of the page limit set for System/Demo papers (unless this is another type of paper, which should then be made more clear),
* fix the grammar.

Because of this I lean slightly towards reject.

Below I've collected a few additional remarks per section:

**3.1 Datatype map (input 5)**

> In the case of input 5, it is intended that mapping languages would be able to generate datatype tags according to the most probable value according to values formats. For example, in input 5 it is expected that number 3 would have an xsd:integer datatype and 3.14 would have an xsd:decimal one.

Looking at the input, the value for `num` in the second object is `"3.14"`. Now, I don't know if this was intentional, but in my opinion it highlights a significant problem with the presented approach in ShExML.
If such automatic datatype inferences were to be supported, they should, in my opinion, only infer based on available information in the source document structure. That is to say, if the source document type has certain native datatypes, one could use that information to infer a datatype during mapping.
In the case of the value `"3.14"` in a JSON document, I would definitely expect `"3.14"` to have datatype `xsd:string`, because the JSON datatype is also string. However, in ShExML solution, the datatype would become `xsd:decimal`. The ShExML datatype inference is indeed "naïve", as the author puts it. But, arguably also incorrect, since I believe it will result in incorrect or unintended inferences for users.

About the naivety of the implementation the author states:

> However, it can lead to a more complex inference system if it is desired or needed.

However, there is no further mention on how this can be done. Would this be part of the language? And if so, how?

**3.2 Join on literal**

As a reader unfamiliar with ShExML it is hard to figure out just from a ShExML snippet how the joins work. It would be better to succinctly explain how joins in ShExML work (or provide a reference), and thereby show that this is not a challenge in ShExML.
I feel like this section, in its current form, could be replaced with just a reference to the ShExML specification, and the supplementary webpage showing that ShExML has solved this challenge.

**3.3 Multivalue references**

The author states:

> Therefore, it seems that in hierarchical data the expected output should be a verbatim translation.

It is unclear to me what a "verbatim translation" means in this context, or how the above conclusion is reached, making the rest of the section also unclear to me.

**4.1 Access fields outside iterators**

This section introduces an interesting mechanism to move a field down the hierarchy during iteration, however the explanation is really superficial and lacks clarity. Next to this, from looking at the ShExML snippet, I think the semantics of the words push and pop are strangely applied in this solution. I would expect a pushed field to be pushed down the hierarchy, and a popped field to be popped up a level in the hierarchy, or something in that vein, but that doesn't seem to be the case here. If I understand it correctly, the popped field is not really "popped", but references the "pushed" field, which is not really pushed but saved during a higher iteration for use in its nested iterations.

**5.2 Excel Style**

The author explains how one could generically solve this issue, but it is unclear how this relates to ShExML.

---

### Official Review · ~Antoine_Zimmermann1 · 2021-04-18
**A report on challenges addressed and yet to be addressed by ShExML**

**Rating:** 6
**Confidence:** 4

**Review:**

The paper describes how the language ShExML can address various challenges of data transformation that have been identified in the Knowledge Graph Construction community group. It reports on what was possible in the early versions of the language, what changes were made to address more challenges, and what's still remaining to be solved.

This paper could be useful to people who are using ShExML when faced with similar challenges as those identified by the community group, or people who would like to implement a solution for mapping data to RDF such that the solution addresses the challenges. However, as a research paper, it does not bring much.

First, I would argues that the "research questions" provided at the beginning are more engineering problems than true research. They are similar to "how to implement X in language Y". For instance, "how to implement a Web server in pure SQL"? This is indeed a challenge but hardly a research problem. The research question should rather be something like "how to address the challenge X with requirement Y" and possibly requirement Y directs the solution towards using a certain language, due to certain characteristics of the language.

Second, even if this reports on ShExML, its evolution and implementation, it would have been good to provide comparison with other languages. Can RML address the same challenges? To what extent? How complicated are equivalent mappings in RML? What about SPARQL-Generate, which is somewhat similar in its syntax to ShExML?


More specific comments:
In Table 1, it is not clear what the last column means.
In Listing 1.1, why not include a conversion to dates and datetimes?
In Listing 1.2, the join on firstname seams strange when the goal is to get family names


Smaller issues:
Intro: "How can not addressed challenges" -> How can unaddressed challenges
Sec.2: "how not solved challenges" -> how unsolved challenges
Sec.3: "those which are reachable with" -> those that are reachable with
Sec.3.3: "This case comes more complicated" -> This case becomes more complicated (?)
Sec.4.1: "need to obtain values which are parents" -> need to obtain values that are parents
Sec.4.4: "mak it as simpler as possible" -> make it as simple as possible
Sec.5: "to rethought" -> to rethink
Sec.5.2: "to convert them to CSV a then treat" -> to convert them to CSV then treat
Sec.6: "continuist" -> is this a real word? I could not find it in several dictionaries
Ref: [6] "Lefrancois" -> "Lefrançois"
Ref: [8] "fno" "dbpedia" -> capital letters needed
Ref: [10] and [11] "xr2rml" -> xR2RML

---

### Official Review · ~Thomas_Delva1 · 2021-04-21
**Overview of additions to ShExML guided by the KGC challenges**

**Rating:** 7
**Confidence:** 5

**Review:**

The author goes over the mapping challenges proposed by the W3C KGC working group and for each discusses support in ShExML.
The mapping challenges are categorized based on whether they were already supported in ShExML, whether support was added as a paper contribution, or whether they remain unsupported by ShExML.
For each challenge it is also discussed how other mapping languages might deal with this type of challenge.
For the still unsupported challenges this discussion leads to perspectives on how to add support for these challenges in the future.

The paper presents a nice set of additions to the ShExML language, making the language more mature and complete.
The quality of the presented solutions is high overall, some minor comments are given below.
The solutions are framed well with relation to existing languages which offer similar solutions.
It would have been interesting to give a more structured comparison of how different languages tackle the challenges,
but that was not the goal of the paper (as the title already indicates).

I have following comments on the content which might need to be addressed to improve the work's quality:

* In section 3.1 a datatype inferencing mechanism is presented. The mechanism is based on trying to parsing strings as different datatypes and returning the first succesful datatype from a list. While not bad per se and obviously useful in many cases, this mechanism has one drawback: it first turns the data value into a string. Many data models have their own internal data types, e.g., JSON has strings, booleans, numbers. By first turning all values that have such a data type into strings, that information is lost. In contrast, R2RML has a section on converting SQL data types into corresponding xsd types: https://www.w3.org/TR/r2rml/#datatype-conversions . A good solution would be to extend these conversions to other data models than the SQL2008/relational model.

* On page 5 it is mentioned that "to generate the cartesian product it would be as easy as to define different top level iterators".
This idea is clear, however no example is included or attached which makes hard to understand how it would look like in ShExML.
Adding an example (maybe on the external webpage with the other examples) would make things more clear.
I tried to make my own example but could not immediately get it working, so I do not think it is trivial.

* In section 4.1 it is not entirely clear why pushed and popped fields are needed. Consider this slightly modified example:

```
ITERATOR records <jsonpath: $.records[*]> {
    FIELD id <id>
    ITERATOR cars <cars[*]> {
        FIELD make <make>
    }
}

ex:Car exPerson:[input.records.cars.make] {
  	ex:owner exPerson:[input.records.id] ;
}
```

It seems like the engine can effortlessly determine from the fields' names that `input.records.id` comes from one level higher than `input.records.cars.make`, so the first needs to be pushed down.
Letting the engine determine this would relieve users from declaring the pushes and pops themselves.
For context, in xR2RML which introduced pushed/popped values determining this is not possible, since there references have no name which shows which level they are on.

Is there an implicit assumption all fields in a shape should come from the same hierarchical level?
Does letting the user declare pushing/popping give them more fine-grained control?
These things should I think be clarified.

* In section 4.5 a syntax is presented for generating RDF collections/containers. The design of this syntax seems counterintuitive: RDF containers are *around* RDF terms, but in this language their definition is *inside* a term, e.g., `exPerson:[article.author.name AS  RDFList]`. This makes it seem like `exPerson:` is not part of the collection, while it is.
Perhaps clarify why this design choice was made over the more intuitive `exPerson:[article.author.name] AS RDFList` (or something similar).

Further, these comments on details of style would improve the paper if addressed:

* Since the paper relies partly on externally hosted content, using clickable hyperlinks would make it more readable.

* When describing the contents of sections after the introduction, a description of section 2 is missing.

* Table 1 is hard to interpret: it contains checkmarks and exceptions (subchallenges) between parentheses, but sometimes these exceptions are positive and sometimes negative. Since the paper is well under the page limit, perhaps consider adding a row in the table for each subchallenge.

* In the first demo under "Generate multiple values", the user interface gives a warning on parsing `FIELD lastname <firstname[1].label>`.
Probably numbers need to be added to the allowed characters.

* A similar warning is given on input 4 of "Datatype map" for (I believe?) double quotes inside matcher expressions.

* There are some language mistakes in the text. E.g., p1 "complicate" should be "complicated"/"complex".

---

### Meta-Review · Program_Chairs · 2021-04-21

**Recommendation:** Accept
**Confidence:** 5

**Metareview:**

The four reviewers agreed on the relevance of the presented work for the paper. We would encourage the author to look into the comments provided (e.g., research questions vs engineering problems) and include them in the camera-ready version of the paper. Consider that the paper can be extended until 15 pages. For sure, this work will generate useful and interesting discussions during its presentation in the workshop.

David

---

### Decision · Program_Chairs · 2021-04-23

Accept